# A Narrative Review of Human Clinical Trials to Improve Lactose Digestion and Tolerance by Feeding Bifidobacteria or Galacto-Oligosacharides

**DOI:** 10.3390/nu15163559

**Published:** 2023-08-12

**Authors:** Sindusha Mysore Saiprasad, Olivia Grace Moreno, Dennis A. Savaiano

**Affiliations:** Department of Nutrition Science, College of Health and Human Sciences, Purdue University, West Lafayette, IN 47907, USA; omoreno@purdue.edu

**Keywords:** probiotics, prebiotics, *Bifidobacterium*, galacto-oligosaccharides, lactose intolerance

## Abstract

Supplementation with the probiotic *Bifidobacterium* and prebiotic galacto-oligosaccharides (GOS) could improve gut health and benefit lactose intolerant individuals. A narrative review was conducted to identify human clinical trials that evaluated lactose digestion and/or tolerance in response to consumption of *Bifidobacterium*, GOS, or both. A total of 152 studies on *Bifidobacterium* and GOS or both were published between 1983 and 2022. Out of the 152 studies, 20 were human clinical trials conducted in lactose intolerant subjects; 8 studies were conducted with *Bifidobacterium* supplementation and 3 studies involved GOS supplementation. Five studies reported favorable outcomes of *Bifidobacterium* supplementation in managing lactose intolerance (LI). Similarly, three studies reported favorable outcomes with GOS supplementation. The other three studies reported neutral outcomes. In conclusion, most studies reported a favorable effect of *Bifidobacterium* and GOS on managing the symptoms of LI. No study has examined the effects of combined supplementation with *Bifidobacterium* and GOS in lactose intolerant subjects. Future research could examine if co-supplementation with *Bifidobacterium* and GOS is a more effective strategy to reduce the dairy discomfort in LI individuals.

## 1. Introduction

Lactose intolerance (LI) is often managed with dairy restriction or lactase supplements. Certain intestinal micro-organisms including *Lactobacillus* and *Bifidobacterium* reduce the lactose concentration through their β-galactosidase activity [1]. Altering the gut microbiota could be a long-term strategy to manage lactose intolerance. We reviewed research which aims to improve lactose digestion and tolerance by feeding *Bifidobacterium* and/or galacto-oligosaccharide (GOS) to modify the colonic microflora and alter the fermentation of undigested lactose. Consuming probiotics and prebiotics to modify the microbiome has been established [1]. *Bifidobacterium* can efficiently utilize GOS as a substrate for growth [2]. A metagenomic study by Blekhman et al. [3] suggests that *Bifidobacterium* is present in abundance in the lactase non-persistent population. Further, the β-galactosidase activity of *Bifidobacterium* enables these individuals to digest lactose more efficiently [4]. Given the selective influence of GOS on *Bifidobacterium*, feeding both *Bifidobacterium* and GOS could be an efficient strategy to manage lactose intolerance. However, there are limited studies exploring the combined potential of these probiotics and prebiotics in managing LI. The studies reviewed identify research gaps and prioritize further research to improve lactose tolerance.

## 2. Materials and Methods

We searched PubMed, Scopus, and Web of Science for original human clinical trials that fed *Bifidobacterium*, GOS, or both and evaluated lactose digestion and/or tolerance. Search terms included probiotics, prebiotics, and lactose intolerance. The search terms used are described in Figure 1. Interventional, randomized-controlled trials (RCTs) with or without placebo and a control group were reviewed. Studies identifying possible outcomes, i.e., positive, neutral, or negative outcomes, with *Bifidobacterium*, GOS, or their combination and lactose intolerance were considered. A decrease in hydrogen breath levels (indicating improved lactose digestion), abdominal discomfort scores, other LI symptom improvement, and an increase in *Bifidobacterium* in the feces were considered as positive outcomes [5,6,7]. Increased intolerance symptoms, HBT levels, and fewer *Bifidobacterium* in the feces were considered negative outcomes. No change in the outcomes was considered neutral. Mixed results, such as a decrease in symptoms or tolerance in one phase of the study or with a specific combination of interventions, were also considered under the neutral outcomes category. We searched English language studies with no restriction on year published or age of participants. *Bifidobacterium* was the primary probiotic of interest. Studies with other probiotics were included if they were fed with *Bifidobacterium*. The inclusion and the exclusion criteria are described in Table 1.

A preliminary search was conducted to identify human clinical trials. For the preliminary search, the title, abstract, and the keywords were reviewed. Articles that provided full text and were human clinical trials that assessed lactose intolerance symptoms, lactose maldigestion, and the microbiome were reviewed. Finally, the randomized controlled trials among the selected papers were screening using Crochane’s test of bias [8].

## 3. Results

The search for *Bifidobacterium*, GOS, probiotics, prebiotics, and LI returned a total of 152 studies published between 1983 and 2022. Among these 152 studies, most evaluated the role of *Bifidobacterium*/probiotics and/or GOS/prebiotics in other health conditions (Figure 2). Eight studies evaluated the effect of consuming *Bifidobacterium* with or without other probiotics on people with lactose intolerance [4,9,10,11,12,13,14,15] (Table 2). Only three studies evaluated the efficacy of GOS in people with lactose intolerance [16,17,18].

In total, eleven studies selected for review tested the efficacy of *Bifidobacterium* and/or GOS to improve lactose tolerance [4,9,10,11,12,13,14,15,16,17,18] (Table 3). Eight of these studies reported a significant improvement in lactose digestion and/or tolerance [4,10,11,12,15,16,17,18], while three reported neutral outcomes [9,13,14]. All the studies were at a low risk of bias as per the Cochrane’s analysis.

### 3.1. Bifidobacterium as a Digestive Aid for People with Lactose Intolerance

Of the eight studies that fed *Bifidobacterium*, three were acute studies that fed the probiotic for a week or less [9,10,15] and five were chronic feeding studies that fed the probiotic for more than a week [4,11,12,13,14]. All eight studies evaluated if *Bifidobacterium* supplementation reduced symptoms of lactose intolerance such as bloating, constipation, abdominal pain, and diarrhea [4,9,10,11,12,13,14,15]. Changes in the gut microbiome were reported in three studies by analyzing the fecal metabolite concentration and the β-galactosidase activity [4,12,14]. Participants self-declared a decrease in intolerance symptoms post-intervention in five studies [4,7,8,9,12], while three reported neutral results [9,13,14].

*Bifidobacterium* was supplemented alone in four studies [4,9,14,15]. The other four studies fed additional probiotics, mostly *Lactobacillus* [10,11,12,13]. In these studies, the efficacy of *Bifidobacterium* and the other probiotics was tested in reference to placebo or with lactase-supplemented control groups.

In one double-blind, randomized, cross-over, placebo-controlled study conducted by Vitellio et al. [12] however, the efficacy of *Bifidobacterium* was tested along with vitamin B6. *B. longum* BB 536, *L. rhamnosus* HN001, and vitamin B6 were fed to 23 lactose intolerant participants. The participants who reported persistent symptoms of LI despite being on a lactose-free diet for 6 months were enrolled in the study. Supplementation with probiotics and vitamin B6 for 30 days alleviated gastrointestinal distress symptoms among the participants when on a lactose-free diet. The authors also reported an elevated population of *Bifidobacterium* in the fecal microbiome, while the levels of *Lactobacillus* remained consistent across both groups. Vitellio et al. [12]. reported a correlation between the fecal *Bifidobacterium* and LI symptoms. The treatment group reported reduced bloating and constipation, and an increase in resident *Bifidobacterium* was found. Abdominal pain and intestinal permeability were not affected. According to the Bristol scale, sub-analysis showed a trend towards normality after treatment. Interestingly, despite being on a low-carbohydrate diet through the intervention, supplementation with *Bifidobacterium* and *Lactobacillus* reduced the incidence of constipation in the experimental group.

*Bifidobacterium* was supplemented with *Lactobacillus* in four studies [10,11,12,13]. In a one-week intervention, Masoumi et al. [10]. fed yogurt fortified with *Bifidobacterium* sp. and *L. acidophilus* or a placebo to people with lactose intolerance. The control group was supplemented with *L. bulgaricus* and *S. thermophiles* (typical yogurt-making bacteria that do not survive in the gastrointestinal tract), while the experimental group received *L. acidophilus*, *Bifidobacterium*, and *S. thermophiles*. Hydrogen breath levels were measured before and after the supplementation. Subjects consumed yogurt three times a day for one week. The group receiving the probiotic yogurt with *Bifidobacterium* sp. and *L. acidophilus* showed improved lactose digestion (decrease in HBT) after the intervention. Further, bloating and flatulence improved significantly in the experimental group. Abdominal pain and nausea did not change. Sustained lowering of LI symptoms several weeks post-study was significant, suggesting a long-term alteration of lactose fermentation in the colon.

Almeida et al. [11]. fed probiotics, *L. casein*, and *B. breve* [powder dissolved in water] to 27 subjects for 4 weeks. Almeida et al. [11]. were attempting to evaluate the long-term effects of probiotic supplementation on managing lactose intolerance. The extended intake of probiotics reduced symptoms of lactose intolerance. Probiotic supplementation caused a significant decrease in the abdominal pain score, diarrhea, and bowel frequency post-lactose load. Lactose digestion was assessed by HBT after a lactose load during each test visit. In the probiotic-fed group, symptoms and breath hydrogen decreased from baseline after treatment and remained low after 3 months. Not surprisingly, this reduction was not as great as that in a lactase-supplemented group. Although the symptom scores for the probiotic supplementation group did decrease, the hydrogen breath test score was still higher for probiotic use than in the lactase-fed group. Further, the participants were on a milk-free diet during the probiotic intervention period. The absence of a lactose-rich diet during supplementation makes it challenging to determine the effectiveness of these probiotics in maintaining microbial modifications. Perhaps, continued dairy consumption prolongs the positive effect of the probiotic on fermentation. Given that we know that lactose is a prebiotic, this hypothesis seems likely.

While Masoumi et al. [10] and Almeida et al. [11] individually tested the acute and chronic consumption of *Bifidobacterium*, Aguilera et al. [9] tested both acute and chronic ingestion of *B. Bifidum* 900791 at high and low concentrations in lactose maldigesters in a single study. During both phases, participants completed HBTs, each one a week apart, to measure lactose digestion after the consumption of ice-cream without *Bifidobacterium*, with either a low, high probiotic concentration, or with lactase, in a randomized order. Digestion of ice-cream with either the low or high dose of probiotic improved lactose digestion following a single meal to a level similar to consumption of the dairy product with lactase. Additionally, abdominal pain was mitigated after the consumption of probiotic ice-cream to a level similar to that caused by ice-cream with exogenous lactase. Further, there was no difference in abdominal pain between high- and low-level probiotic consumption. The chronic phase of the Aguilera et al. [9] study required participants to consume either a low concentration of *Bifidobacterium* or a placebo daily for 4 weeks. After the chronic phase, a final HBT was conducted with *Bifidobacterium*-free ice-cream. When compared with baseline HBT, no difference was reported and there was no decrease in symptoms. This contradicts the results of Almeida et al. [11], where chronic ingestion of *Bifidobacterium* kept the HBT levels reduced even three months after the supplementation. Aguilera et al. [9] suggested that the difference in results could be due to unmonitored dietary patterns or the small sample size.

To understand if supplementation with *Bifidobacterium* could be as effective as lactase, Rasinkangas et al. [14] conducted cross-over clinical human trials feeding milk or lactose in water as a lactose source. Lactose intolerant individuals were either supplemented with Bi-07, 4662 FCC lactase, or placebo [14]. The Bi-07 produced high lactase activity in the feces. The authors observed variable results depending on the source of lactose and the duration of the study, likely due to differences in the protocols. The observed increased efficacy of Bi-07 in comparison with the placebo could be due to the prolonged intake of the probiotic. Further, a carry-over effect was observed when the participants shifted to placebo after probiotic supplementation. Despite an increase in the fecal β-galactosidase activity, Bi-07 did not always reduce symptoms. The inconsistent results could be attributed to the carry-over and sequence effects.

The β-galactosidase activity of *Bifidobacterium* in reducing the symptoms of LI appears to be dependent on the bacterial strain and its cultured environment. Tianan et al. [15] fed unfermented milk containing *B. longum* to 15 individuals with lactose malabsorption. The study supplemented two strains of bacteria, *B. longum B6* and *B. longum* ATCC 15708, with a non-probiotic control. Both *Bifidobacterium* strains were grown in a lactose-rich environment and fed to the participants in non-fermented milk. To further understand the influence of growth medium on the bacteria’s metabolic activity, the participants were also fed a *B. longum* strain cultured in lactose or lactose and glucose. Beta-galactosidase activity was higher in the strains grown on lactose compared with those cultured on lactose plus glucose. Tianan et al. [15] suggest that both growth media and strain play a role in the effectiveness of *Bifidobacterium* in decreasing LI symptoms.

The reduction in the LI symptom score with *Bifidobacterium* supplementation is likely attributed to a shift in the gut microbiota population. When He et al. [4] supplemented 11 lactose intolerant participants with *Bifidobacterium*, a change in the microbiome was observed. During this study, He et al. [4] wanted to understand if simultaneous ingestion of *B. longum* (as a capsule) and *B. animalis* (in yogurt) for 2 weeks can modify the microbiota of the colon. Fecal samples show that the supplementation favored the growth of the two provided strains of the bacteria, although not significantly. Additionally, PCR-denaturing gradient gel electrophoresis showed that *B. animalis* and *B. longum*, which gained residence in the gut during supplementation, disappeared when the intervention stopped. The supplementation alleviated the lactose intolerance symptoms during the study (measured by diarrhea). However, these effects also lasted only during the period of supplementation. While symptoms of LI decreased, the degree of lactose digestion remained constant and, hence, researchers estimated that the brush border enzymes were not affected by consumption of the bacteria. Beta-galactosidase activity also increased after supplementation, but not to a significant amount. This could be because it is harder to change the microbiota of adults than younger participants. It is important to note that the sample population (people of Chinese heritage) have naturally lower lactase activity than other potential sample populations. Due to a small sample size, the study should be repeated with a larger sample size to avoid sampling bias. Nevertheless, the reduction in the symptoms observed during the supplementation highlights the role of *Bifidobacterium* in managing lactose intolerance.

While most of the studies report a positive effect of *Bifidobacterium* supplementation and lactose tolerance, a study by Roškar et al. [13] observed mixed results. In this study, the 44 participants received either a probiotic or placebo for 6 weeks. *B. animalis* and *L. plantarum* MP2026 were selected due to their resistance towards acidic environments and their ability to digest lactose. The selected probiotics were distributed in capsule form and were consumed two times daily for 6 weeks. HBT was performed and symptoms were measured at baseline, after six weeks, and then a 2-week follow-up. During the study, both groups reported a decrease in symptoms, which suggests a placebo effect and, thus, made it more challenging to determine if the probiotics were a therapeutic agent in managing lactose intolerance. However, the study did report a decrease in the diarrhea and flatulence scores. The suspicion was resolved when probiotic consumption lowered LI symptoms in only the treatment group after the follow-up, thus suggesting that the probiotic was effective. It is important to note that there was little variability in HBT results for the treatment group, suggesting the study needs to be repeated. These variable outcomes of Roškar et al. [13] could be attributed to the uncontrolled dietary patterns of the participants during the intervention. To obtain more reliable results, further studies could focus on a larger sample size and applying similar dietary restrictions to all participants.

### 3.2. Galacto-Oligosaccharides as a Digestive Aid for People with Lactose Intolerance

Two large clinical trials examined the effects of galacto-oligosaccharide consumption alone in individuals with lactose intolerance [16,18]. Further, one follow-up study analyzed the effect of GOS on the microbiome [17]. While the search results for galacto-oligosaccharides identified 42 studies, most were animal studies. A double-blind, placebo-controlled study by Savaiano et al. [16] studied if alteration of the gut microbiome by consumption of RP-G28 (a GOS) could improve lactose digestion and tolerance. Savaiano et al. [16] administered RP-G28 to 85 lactose intolerance participants for 35 days, with the load increasing every 5 days. The intervention increased the lactose tolerance in the participants and significantly reduced the abdominal pain. Lactose digestion improved and symptoms of LI decreased after treatment with GOS as compared with placebo. Dairy was re-introduced after 30 days and the participants’ tolerance to dairy was measured through HBT and symptom reports. Nearly 80% of GOS consumers claimed they no longer experienced abdominal pain.

Azcarate-Peril et al. [17] evaluated if an increased intake of RP-G28 GOS altered the microbiome population in the gut. In this study, Azcarate-Peril et al. [17] collected fecal samples prior to, during, and after consumption of RP-G28. Nearly 90% of the participants showed an increase in the lactose-fermenting microbiome population in the stool samples. The *Bifidobacterium* genus increased in 90% of consumers on day 36; however, on day 66 [after consumption of GOS had ceased], levels had returned to baseline. The shift correlated with an improvement in lactose intolerance symptoms. This positive correlation of *Bifidobacterium* and GOS suggests that GOS impacts LI symptoms by increasing the bifidobacterial count.

A follow-up large-scale study conducted by Chey et al. [18] on 377 lactose malabsorbers not only observed a decrease in the abdominal pain score, but also an increase in the quality of life after the RP-G28 intervention. In this study, Chey et al. [18] fed two different doses of RP-G28 and compared the effects on LI symptoms with a placebo [18]. The participants in this study were administered RP-G28 or a placebo for 30 days and milk consumption was continued for another 31 days. Each subject was randomly assigned either the lower dose (5 g increased on day 10 to 7.5 g) or the higher dose (7.5 g increased on day 10 to 10 g) to consume in packets two times daily. Results showed a significant decrease in cramping and bloating for treatment. Subjects who consumed the GOS also reported drinking more milk on average post-study than subjects who received the placebo. Chey et al. [18] also observed that the alteration in tolerance levels corresponded to an increase in the lactose-fermenting microbes. There was an increase in *Bifidobacterium* and a decrease in breath hydrogen in the treatment group. However, the article did not explain any differences in the varying concentrations of RP-G28 doses.

## 4. Discussion

*Bifidobacterium* could be an effective approach to managing LI symptoms. *Bifidobacterium* supplementation for a week or less [9,10,15] decreased abdominal distress symptoms in lactose intolerant participants. The reduction in pain was sustained even when supplementation was discontinued in a few studies, but this finding was not uniform.

While *Bifidobacterium* and GOS were efficient in reducing lactose intolerance symptoms in most studies, the specific symptom improvement was somewhat variable. The symptoms that decreased after supplementation included abdominal pain, diarrhea, and bloating [11,12,18]. Several studies reported an overall decrease in symptoms [4,10,13,16]. The neutral effects in a few studies could be attributed to the carry-over, sequence effects, or the duration of the study [9,13,14]. In most studies, Bifidobacterium was efficient in reducing the abdominal symptom scores when compared with placebo. Compared with lactase, *Bifidobacterium* was less efficient. When Aguilera et al. [9] tested the efficacy of chronic consumption of *Bifidobacterium*, there were significant differences in the abdominal pain score. However, it is not surprising that Aguilera et al. [9] reported no significant decrease in other symptoms compared to the control group, since the control group was supplemented with lactase.

Lactose digestion as measured by the HBT was a common method for these studies. A decrease in HBT levels correlated with a decrease in digestive symptoms as well. Abdominal discomfort scores after a decrease in HBT levels was evident in the acute phase of the study conducted by Aguilera et al. [9]. Similarly, a decrease in the breath levels in the study by Masoumi et al. [10] correlated with a decrease in symptoms. It comes as no surprise that when HBT levels remained unchanged, the scores of the abdominal symptoms did not alter much, either.

Another question to consider is if co-colonization of other probiotics with *Bifidobacterium* is beneficial. Almeida et al. [11], Masoumi et al. [10], and Roškar et al. [13] supplemented *Bifidobacterium* with other bacteria. In these studies, symptom scores remained low for a few weeks after the intervention. In the study by Almeida et al. [11], *B. breve* in addition to *L. casein* reduced the symptoms of LI as effectively as lactase supplementation. Supplementation with *B. breve* with *L. casein* resulted in a lower HBT, even 3 months after the intervention. On the other hand, the effects were short-term in the studies that supplemented *Bifidobacterium* alone [4,9,15]. Vitellio et al. [12] and Masoumi et al. [10] reported a definite improvement in LI symptoms in presence of *Bifidobacterium*. Considering this improved efficacy of *Bifidobacterium* in the presence of *Lactobacillus*, further studies could explore the potential of other probiotics as supplementary interventions with *Bifidobacterium*.

When combining *L. acidophilus* with *Bifidobacterium*, determining the effective dose and duration of supplementation with probiotics may be a challenge. The efficacy of *Lactobacillus*, in particular, seems to seem to be strongly correlated to its concentration. In a study by Lin et al. [19], *L. acidophilus* in 10^8^ and 10^9^ CFU/mL did not cause a significant decrease in hydrogen levels. However, Kim et al. [20] observed that a 10^8^ CFU/mL concentration of *L. acidophilus* caused a significant decrease in maldigestion. These studies also differed in the duration of supplementation [20]. While the Lin et al. [19] intervention was for a single day, Kim et al. [20] supplemented the probiotic for 6 days. Hence, while supplementing *L. acidophilus* with *Bifidobacterium*, determining the optimal dose and the adequate period of supplementation should be evaluated for a clinically relevant improvement in lactose digestion.

The evidence comparing the efficacy of acute versus chronic consumption of *Bifidobacterium* is not clear. While Almeida et al. [11] reported a persistent effect of *Bifidobacterium* post-supplementation, the results of Aguilera et al. [11] were contradictory. This could be because of the large number of dropouts in the study of Aguilera et al. [9] and/or poor dietary control, which may have included the effect of dietary fiber in altering the H2 levels. More studies are needed to evaluate the time necessary to provide long-term improvement of lactose digestion and tolerance using *Bifidobacterium*.

GOS was effective in managing LI. Azcarate-Peril et al. [17] provide evidence that GOS as a supplement can significantly increase *Bifidobacterium* populations in the large intestine. Consistent with this finding, Davis et al. [21] found that consumption of GOS could selectively increase *Bifidobacterium* in the gut microbiome at the cost of bacteroides. Davis et al. [21] examined stool from adults who consumed GOS over 12 weeks. Further, *Bifidobacterium* strains have been evaluated for their variability in digesting GOS. When comparing 14 *Bifidobacterium* strains in the presence of GOS in vitro, *B. adolescentis*, *B. catenulatum*, *B. pseudocatenulatum*, and *B. infantis* all showed the greatest increased density and most rapid GOS consumption compared with other strains [22].

### Limitations

An inconsistency in post-intervention diets is one of the limitations of the studies in this review. Controlling the intake of dairy lactose and fiber, while difficult and expensive, would likely give more clarity to the long-term efficacy of *Bifidobacterium* and/or GOS. Further, cross-over studies with a significant wash-out period might reduce some of the observed variability. Small sample sizes are another common limitation of these studies.

GOS reduced the symptoms of dairy intolerance following both acute and chronic ingestion. However, there were few studies evaluating the effect of GOS on lactose digestion and tolerance, and they were all conducted by the same group. GOS promoted the growth of the *Bifidobacterium* and decreased HBT levels and abdominal discomfort [16,17]. Supplementation with GOS caused a long-term bifidogenic response. Studies that simultaneously test the individual effects of *Bifidobacterium*, GOS, and the combined effect of both under identical controlled conditions could help identify a more effective approach for managing LI.

## 5. Conclusions

When fed individually, both *Bifidobacterium* and GOS have been shown to improve lactose digestion and/or tolerance. Further, GOS stimulates the gut microbiota population in favor of *Bifidobacterium*. Although in some studies the decrease in discomfort was not as large as complete dairy avoidances, symptoms did ameliorate significantly compared with dairy or lactose-matched controls. Further, supplementation did not cause any significant side-effects. People with lactose intolerance could reap benefits from a positive shift in the microbiota that use lactose as a substrate for growth. Co-supplementation with *Bifidobacterium* and GOS could be an even more effective management strategy to reduce the dairy discomfort in LI individuals.

## Figures and Tables

**Figure 1 nutrients-15-03559-f001:**
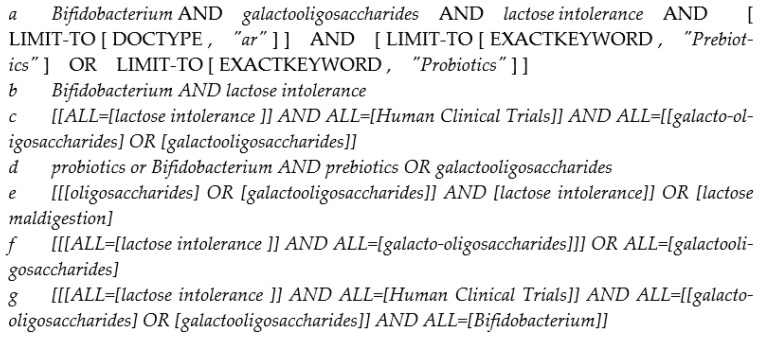
Keywords used.

**Figure 2 nutrients-15-03559-f002:**
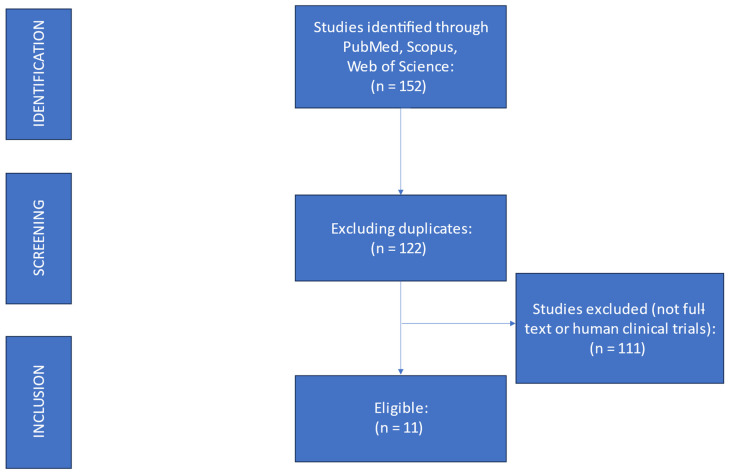
Flow diagram of the search process.

**Table 1 nutrients-15-03559-t001:** Inclusion and exclusion criteria.

Parameter	Inclusion Criterion	Exclusion Criteria
Participants	No age limit	Gastrointestinal disorder other than LI, antibiotic treatments
Intervention	*Bifidobacterium* or galacto-oligosaccharides	-
Outcomes	Reduced hydrogen breath test [HBT] results, decreased abdominal discomfort scores, reduced fecal urgency, and improved lactose tolerance	-
Study design	Human clinical trials, randomized controlled trials (RCT’s)	In vivo and in vitro experiments

**Table 2 nutrients-15-03559-t002:** Summary of study outcomes.

Variable Studied	Total Number of Studies	Outcomes Studied	Number of Studies with Favorable Outcomes	Number of Studies with No Significant Outcomes	Number of Studies with No Favorable Outcomes
Lactose intolerance and *Bifidobacterium*	74	8	5	3	0
Lactose intolerance and Galacto-oligosaccharide	48	3	3	0	0

**Table 3 nutrients-15-03559-t003:** Summary of studies.

S. No.	Ref No.	Subjects			Intervention		Days of Intervention	Symptoms Evaluated	Hydrogen Breath Test (HBT)	Fecal Analysis
		N	Control/Placebo	Lactose Maldigesters	Control/Placebo	Experimental Group				
1	[4]	11	0	11	None	*B. longum* and *B. animalis*	2 weeks	All symptoms of LI	Conducted during each visit	Analyzed for fecal microbiota and β-galactosidase activity
2	[9]	45	Acute phase = 45 Chronic phase = 16	Acute phase = 45 Chronic phase = 13	No *Bifidobacterium* (negative control)/lactase (positive control)	*B. bifidum* 900,791	Acute phase: same-day analysis Chronic phase: 4 weeks of placebo/*Bifidobacterium*	Nausea, vomiting, abdominal pain, abdominal distension, increased rectal gas, borborygmi, and diarrhea	Conducted during each visit	None
3	[10]	55	28	27	*L. bulgaricus* and *S. thermophiles*	*L. acidophilus* and *Bifidobacterium* sp.	1 week	Diarrhea, abdominal pain, flatulence, vomiting and nausea, bloating, and flatulence	Conducted during each visit	None
4	[11]	27	27	27	Lactase ingestion	*L. casei* and *B. breve*	4 weeks	Abdominal pain, flatulence, stool consistency, and abdominal distention	Conducted during each visit	None
5	[12]	23	23	23	Placebo with maltodextrins	*B. longum* BB536 + *L. rhamnosus* HN001 + vitamin B6 (ZR)	30 days	Bloating, abdominal pain, and bowel movements	For screening	Analyzed to identify the bacterial microbiome
6	[13]	44	22	22	Placebo	*B. animalis* IM386 and *L. plantarum* MP2026	6 weeks	Diarrhea, abdominal pain, vomiting, and flatulence or rumble	Conducted during each visit	None
7	[14]	34 (BoosterAlpha) + 33 (Booster Omega)	34 (Booster Alpha) + 33 (Booster Omega)	34 (Booster Alpha) + 33 (Booster Omega)	Booster alpha: (placebo) milk Booster Omega: (control) lactose + water	*B. animalis* subsp. *lactis* Bi-07	Booster alpha—101 days Booster omega—195 days	Bowel movements, vomiting, and stool consistency	Conducted during each visit	For quantification of Bifidobacterium in feces
8	[15]	15	15	15	Low-fat milk with no probiotic	*B. longum B6* grown on lactose/*B. longum B6* grown on lactose + glucose/*B. longum* ATCC 15,708 grown on lactose medium	15 days	Abdominal pain, diarrhea, flatulence, bloating, and abdominal rumbling	Conducted during each visit	None
14	[16]	61	19	42	Placebo (corn syrup)	RP-G28	35 days	Abdominal pain, diarrhea, flatulence, bloating and cramping, symptoms reduced post-RP-G28 intervention	Conducted during each visit	None
15	[17]	368	121	Lower dose of RP-G28 = 126 Higher dose of RP-G28 = 121	Placebo (powdered corn syrup)	Low GOS (10–15 g/day) and high GOS treatments (15–20 g/day)	30 days	Analyzed for screening	Conducted for screening	Collected to analyze the fecal microbiota
16	[18]	368	121	Lower dose of RP-G28 = 126 Higher dose of RP-G28 = 121	Placebo (powdered corn syrup)	Low GOS (10–15 g/day) and high GOS treatments (15–20 g/day)	30 days	Abdominal pain, cramping, bloating, and gas movement	Conducted during each visit	Collected to analyze the fecal microbiota

## Data Availability

No new data were created or analyzed in this study. Data sharing is not applicable to this article.

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
