# Peer review of "A Narrative Review of Human Clinical Trials to Improve Lactose Digestion and Tolerance by Feeding Bifidobacteria or Galacto-Oligosacharides"

_nutrients, 2023, doi:10.3390/nu15163559_

Round 1

Reviewer 1 Report

Authors of the manuscript ” A Narrative review of human clinical trials to improve lactosedigestion and tolerance by feeding Bifidobacteria or Galactooligosacharides” wanted to survey the peer reviewed literature on the use of Bifidobacteria or Galactooligosacharides for improving gut health in lactose intolerant individuals.  The authors do have valid intentions with this manuscript, but needs major improvements in the tidiness and thoroughness of the paper. Much more diligence is needed with this manscript. See the following for major points to be addressed.

*Authors have incorrect use of subscript numbers for affiliations. Please Correct.

*Abstract mentioned probiotic first then prebiotic. For easier reading and clarity, authors should be consistent in which is first (probiotic) and then second (prebiotic) throughout the whole manuscript.

*There are unconventional use of “et al” in the text of the manuscript. Please reword.

Introduction is not extensive.

Methods

*The authors state the following “Studies identifying possible outcomes, i.e., positive, neutral or negative outcomes with Bifidobacterium, GOS, and their combination and lactose intolerance were considered.” But what is meant by outcomes, for example what is meant by a “positive” outcome. Authors must define positive, neutral or negative outcome and what are the outcomes based on. English language is difficult in this sentence too. Please correct.

Figure 1 is not described in the text. It should be. It also needs tidying in its format and why is Keywords used on line 51?

*Authors stated “Studies that included 45 other prebiotics such as Lactobacillus were included if Bifidobacterium was also fed”. This sentence is confusing. What is meant by fed here?

*Authors stated “For the preliminary search, the title, abstract and the keywords were reviewed.”  Why is this a “preliminary search”? What is the proper search then?

*Authors stated “Full texts of all human clinical trials that assessed the lactose intolerance symptoms, lactose maldigestion, and the microbiome were reviewed.” But are there citations that can be provide to describe what the authors are looking for in the text in relation to 1. lactose intolerance symptoms, 2. lactose maldigestion, and 3. the microbiome? Since the authors are specifically looking at these three points, a description or citation is needed to understand this review.

*Table 1 is confusing as it is really not described in the text at all. A clearer and explicit description of inclusion criteria for the studies is needed. For instance line 43 the authors talk about outcomes, positive, neutral and negative, but in Table 1 there is also “Outcomes” defined as “Reduced hydrogen breath test [HBT] results, decreased abdominal discomfort, reduced fecal urgency, and improved lactose tolerance”. There is not citation for these outcomes or reason why the authors are using these criteria. This also applies to the interventions. Why is milk, yoghurt, ice cream, supplement included because the review is about analyzing studies with “Bifidobacteria or Galactooligosacharides”. What is meant with supplement.

*Similarly, authors need explicitly describe the exclusion criteria in the methods section as it is being used to exclude majority of the papers. And be explicit because what is meant by not full-text, etc…?

Results

*Figures 2 is mislabeled as Figure 3.

*Figure 2. What is meant by Studies identified through other sources?  Why are these other sources not mentioned in the methods?

*Generally all the figures and Tables need explicit details in the text and a more informative legend so they can be understood on their own. In addition, authors are really encouraged to read the author’s instructions as there are consistent mistakes in the manuscript, for example square brackets are used for citations and not figures or tables. Please correct.

*In addition, when the authors states 3 studies ….. or 5 studies… .etc,  please proved the exact citation of which studies you are referring to. This must be corrected all through the manuscript.

*It is more inciteful for a review to really analyse the studies. This review generally restates what studies have been done. I would highly recommend that a thorough review and comparison is done on the 11 studies. It is further recommended that PRISMA guidelines are used for this review so that the true impact is demonstrated.

* Authors stated this “Eight of these studies reported a significant improvement of lactose digestion and/or tol-75 erance, while 3 reported neutral outcomes”  But how was the outcomes assessed in Table 2, based on what?

*confusing subheading used, for example why are the authors not using 3.1, 3.2, 3.3

3.1.1. Bifidobacterium as a digestive aid in lactose intolerants 80

3.1.2. Oligosaccharides and digestion in lactose intolerants

3.1.3. Consumption of both Bifidobacterium and GOS in lactose intolerants¨

3.1.3. Consumption of both Bifidobacterium and GOS in lactose intolerants¨

This section is not extensive and really does not add any value to the paper. Please provide a more extensive analysis here.

*”Of the 8 studies that fed Bifidobacterium, 3 were acute and 5 were chronic feeding 81 studies.”  What is meant by acute and chronic feeding?

“Changes in 83 the gut microbiome were reported in 2 studies.” How was this (gut microbiome” assessed?

*Five studies reported decreased intoler-84 ance symptoms from milk, while 3 reported neutral results.”  How were these symptoms assessed, i.e. self declared or? Also see point above in regards to providing the exact citations to the study here.

*A table that compares and contrasts all 11 studies included in this review would provide a more thorough assessment of the studies.

General things

B.longum should have a space between it.

Consistency in writing is lacking in some places. For example, et al or et al. and et al should not be used by convention in the text.

There are many many areas where the text is too vague. Authors should be more explicit with details. For example, “a change in the microbiome was ob-150 served [4]” assessed by how? And “However, these effects also lasted only during the period 158 of supplementation. “ how long was the period?  And Several studies 230 reported an overall decrease in symptoms [4, 6, 9, 13]. What symptoms exactly? Etc.   

Reference

There are numerous mistakes in the references. Please diligently go through all of the references to make sure they are in the proper format as dictated by the journal.

Lastly, the Authors stated the following “

The studies reviewed identify the research 35 gaps and priorities for further research to improve lactose tolerance.”  It would then be beneficial that the authors brings these gaps and priorities out in the discussion part of the manuscript.

Author Response

We thank you for your valuable feedback. We have included the responses in the pdf attached below. 

Reviewer 2 Report

Included in the attachment.

Author Response

We thank you for your valuable feedback. We have added the responses to the comments in the pdf attached below. 
